# The Reuse of Excavated Soils from Construction and Demolition Projects: Limitations and Possibilities

Sarah E. Hale [1,*], Antonio José Roque [2], Gudny Okkenhaug [1,3], Erlend Sørmo [1,3], Thomas Lenoir [4], Christel Carlsson [5], Darya Kupryianchyk [5], Peter Flyhammar [5] and Bojan Žlender [6]

[1] Geotechnics and Environment, Norwegian Geotechnical Institute (NGI), 0855 Oslo, Norway; go@ngi.no (G.O.); ers@ngi.no (E.S.)
[2] Laboratório Nacional de Engenharia Civil, 1700-075 Lisboa, Portugal; aroque@lnec.pt
[3] Faculty of Environmental Sciences and Natural Resource management, Norwegian University of Life Sciences (NMBU), 1433 Ås, Norway
[4] Laboratoire Materiaux pour les Infrastructures de Transport (MIT), Departement Materiaux et Structures (MAST), Université Gustave Eiffel, 44344 Bouguenais, France; thomas.lenoir@univ-eiffel.fr
[5] Swedish Geotechnical Institute (SGI), SE-581 93 Linköping, Sweden; christel.carlsson@swedgeo.se (C.C.); Darya.Kupryianchyk@swedgeo.se (D.K.); peter.flyhammar@swedgeo.se (P.F.)
[6] Faculty of Civil Engineering, Transportation Engineering and Architecture, University of Maribor, 2000 Maribor, Slovenia; bojan.zlender@um.si
* Correspondence: sarah.hale@ngi.no

**Abstract:** The construction and demolition of infrastructure can produce a surplus of excavated soils that ends up at landfills. This practice is not sustainable, and approaches are needed to reduce soil waste and minimize environmental and human health hazards. The "Reuse of urban soils and sites" Working Group in the European Large Geotechnical Institute Platform (ELGIP) works towards a safe and resource efficient use of excavated soils for construction. By considering relevant literature and practicals based on experience in the participating ELGIP countries (France, Norway, Portugal, Slovenia and Sweden), this study presents current practice related to the reuse of excavated soils, and the main barriers (regulatory, organizational, logistical and material quality) to effectively reuse them. Results show that there is no consensus on the best strategies to manage excavated soils in urban areas. This paper provides suggestions of ways in which stakeholders can increase reuse of excavated soils.

**Keywords:** sustainable engineering; excavated soil; reuse; building project; case studies

## 1. Introduction

Current and projected urbanization and concurrent population growth places a greater pressure on natural non-renewable resources and demands a new economic strategy which stimulates the sustainable use of Earth's resources and prevents the production of unwanted waste [1]. Every year, worldwide, 55 billion tons of biomass, fossil energy, metal and minerals are extracted from the Earth and 2.1 billion tons of waste is discarded [2]. Urban centers account for around 80% of carbon dioxide emissions and energy consumption [3,4]. The building and construction sector is responsible for a large part of this consumption and, to meet societal demands, this sector must improve resource efficiency and increase material reuse [5–7].

Building and construction projects (e.g., railways, roads, houses, energy supply networks) often lead to the production of large quantities of excavated soil (both clean and contaminated) which most often ends up at landfill [8]. Current strong signals from the United Nations (UN) and World Health Organization (WHO) have brought sustainable land management into focus. Smarter methods are needed to reduce soil waste and ensure that its reuse does not pose a hazard to public health or the environment. UN Sustainable Development Goal 11: "Make cities and human settlements inclusive, safe, resilient and

sustainable", highlights the fact that sending clean excavated soil to landfill sites is not a sustainable practice. Herein, clean excavated soil is defined as uncontaminated soil, typically removed from the top few meters, with concentrations below relevant threshold concentrations, that originates from construction projects. Reusing excavated soil offers the following benefits: (1) reduction in transportation distance, (2) reduction in costs associated with disposal, (3) preservation of landfill capacity, (4) conservation of mined natural resources, and (5) reduction of environmental and ecological impacts [9].

The surplus excavated soil can be reused on- and off-site, taking into consideration its characteristics and ensuring that these are compatible with the new soil application. Several key geotechnical parameters (e.g., particle size, plasticity, hydraulic conductivity, compressibility, shear strength) and geoenvironmental parameters (e.g., pH, total and leachable concentration of pollutants, organic carbon content) dictate whether soil can be reused in specific situations [10]. In some cases, geotechnical improvement and geoenvironmental treatment of the excavated soil are needed.

Typical on- and off-site options include embankments, roads, railways, landfills, or landscaping. As a rule, on-site reuse is preferred to meet sustainability goals. These management options must be supported by legislation, national guidelines, technical specifications and standards. In Europe, the Waste Framework Directive (WFD, 2008/98/EC) is typically the starting point for the reuse of excavated soils [11]. This directive seeks to " help move the EU closer to a 'recycling society', seeking to avoid waste generation and to use waste as a resource", and specifically states that 70% of construction and demolition waste (CDW), to which excavated soil belongs, should be recycled by 2020. In December 2015, the European Commission (EC) introduced an action plan endorsing the circular economy called "Closing the loop" [12]. Following this, the "EU Construction & Demolition Waste Management Protocol" (European Commission, 2016) and "Guidelines for the waste audits before demolition and renovation works of buildings" (European Commission, 2018) were prepared, however, clean or lightly contaminated excavated soils are not within its scope. In 2020, the EC published a report: "Circular Economy Action plan for a cleaner and more competitive Europe" [13]. Within this report, a new strategy for a sustainable built environment is outlined and one goal is "promoting initiatives to reduce soil sealing, rehabilitate abandoned or contaminated brownfields and increase the safe, sustainable and circular use of excavated soils".

It is clear that the reuse of such excavated soils is a pillar of sustainable geotechnical engineering and there is a growing consensus that achieving a sustainable built environment must start by incorporating sustainability ideas at the planning and design stages of an infrastructure construction project [14,15]. However, so far, only very few countries have developed guidelines or codes of practice to support this (e.g., England/Wales [16], France [17], Australia [18], Canada [19] and Switzerland [20]). These guidance documents provide not only the legal framework for the reuse of excavated soils, but also present best practices for their management, while exercising precaution, particularly in cases where the soil may be contaminated.

Given the imperative need to implement sustainable practices in construction projects supported by legislation and policy makers to increase the reuse of excavated soils, the objective of this work is to present the current status for the reuse of excavated soils, of which the majority are clean or lightly contaminated, in a selection of countries (Australia, Canada, England/Wales, France, Norway, Portugal, Slovenia, Sweden and Switzerland). In this work, such soils are defined as those that exhibit an acceptable risk for public health or the environment (i.e., a low risk) and could be reused directly, with or without geotechnical improvement (but without geoenvironmental treatment, as pollution levels are negligible), for other building and construction purposes. The main barriers to reuse excavated soils are identified and presented in this paper, with the objective to highlight ways that could be used to solve them. Finally, case studies of successful reuse of excavated soils from France, Norway, Portugal, Slovenia and Sweden are documented. These selected countries are members of the Working Group "Reuse of urban soils and sites", which, with other

European countries and respective leading European geotechnical engineering research organizations, constitute the European Large Geotechnical Institutes Platform (ELGIP). ELGIP [21] aims to internationally promote best practice of the profession, its networking and its societal relevance, and its Working Group "Reuse of urban soils and sites" works for a safe and resource efficient reuse of urban soils and sites.

## 2. Current Practices in Several Countries for the Reuse of Excavated Soil

The WFD [11] explicitly refers to clean excavated soils as follows: "The waste status of uncontaminated excavated soils and other naturally occurring material which are used on sites other than the one from which they were excavated should be considered in accordance with the definition of waste and the provisions on by-products or on the end of waste status under this Directive". In addition, the WFD defines a contaminated site as "any soil, water, building or facility that exhibit a level of contamination which is risky for human health or environment", and a waste as "any substance or object which the holder discards or intends or is required to discard". In construction and demolition projects, the starting point for the reuse of soil is to ascertain whether they are classified as waste or not. In some cases (such as in Norway and Sweden), the removal of soil from a site automatically classifies it as a waste. The resultant of this is that large amounts of excavated soil are landfilled (or in some cases, illegally dumped).

### 2.1. Methods Used to Identify Current Practice

An assessment of the methods and policy followed to reuse excavated soil in each country of the ELGIP Working Group was carried out. Information was collected from relevant stakeholders that covered the entire chain of a construction project and included those building on the ground, large companies that had won building contracts and were responsible for design and development (referred to as developers), consultants who were employed at various stages of the construction projects as well as policy makers and enforcers working with excavated soils and construction works. Their opinions were collected during diverse workshops within the ELGIP Working Group in the respective countries via discussion and break out groups. Discussions were focused on what the actors perceived to be the main challenges to increase reuse of excavated soil as well as thoughts related to how these barriers could be overcome. In addition, speakers from countries not included in the ELGIP Working Group were invited to provide insights of how their countries approach reuse of soil from construction projects. Furthermore, ELGIP Working Group members used their personal networks to carry out written correspondence which also focused on barriers to reuse and ways they could be overcome. The opinions expressed in this paper reflect the interpretations of the authors, and not necessarily all of the involved stakeholders, which may present a shortcoming. In addition, it was not possible to ensure that the number of people from each stakeholder group was balanced between countries.

Table 1 provides details of the amounts of excavated soil that is landfilled in the ELGIP Working Group countries. These volumes illustrate the large disparity in amounts of excavated soil produced across countries (France being the largest producer at between 7 and 10,000,000 tons per year, with Portugal the smallest, producing 340,000 in 2017). In addition, they highlight that differences in the management of the reuse of clean soils can dictate the end use and level to which countries reuse soil. For example, in 2018, Norway sent 98% of non-hazardous excavated soil to landfill, which is in contrast to Portugal, who in 2017, sent just 17% of non-hazardous excavated soil to landfill. However, the varying nature of the ELGIP Working Group countries in terms of their social, political and economic status also has bearing on the results.

**Table 1.** Quantities of excavated soil classified as hazardous or non-hazardous waste.

| | FRANCE | NORWAY | | PORTUGAL | | SLOVENIA (Excavated Soil Used On-Site Are Included) | | SWEDEN (Excavated Soil Used On-Site Are not Included) | | |
|---|---|---|---|---|---|---|---|---|---|---|
| **Year** | 2015 | 2017 | 2018 | 2016 | 2017 | 2017 | 2018 | 2012 | 2014 | 2016 |
| Total volume (tons) hazardous and non-hazardous waste | 7,000,000–10,000,000 | 3,010,449 | | 707,810 | 339,386 | 1,938,262 | 3,849,152 | 4,220,000 | 5,690,000 | 5,447,000 |
| **HAZARDOUS WASTE** | | | | | | | | | | |
| Total volume (tons) [6] | | 169,449 | | 12,033 | 14,933 | 4013 | 1917 | 720,000 | 590,000 | 347,000 |
| Backfilling % of total | | | | | | | | | | |
| Recycling [5] % of total | | | | 2768 / 23 | 2987 / 20 | | | 129,000 / 18 [1] | 110,000 / 19 | 111,000 / 32 |
| Pre-treatment [4] % of total | | | | | | 1725 / 43 | 169 / 9 | 285,000 / 40 [2] | 260,000 / 44 | 322,000 / 55 |
| Landfill % of total | | | | 9145 / 76 | 11,946 / 80 | 2288 / 57 | 1748 / 91 | 307,000 / 43 | 220,000 / 37 | 362,000 / 100 |
| **NON-HAZARDOUS WASTE** | | | | | | | | | | |
| Total volume | | 2,841,000 | 2,593,000 | 695,777 | 333,616 | 1,934,249 | 3,847,236 | 3,500,000 | 5,100,000 | 5,100,000 |
| Backfilling % of total | | 585,000 / 21 | 44,000 / 2 | | | | | | 103,000 / 3 [3] | 156,000 / 3 [3] |
| Recycling [5] % of total | | | | 660,988 / 95 | 276,901 / 83 | 52 / 0 | 73,418 / 2 | 103,000 / 3 [3] | 3,300,000 / 65 | 2,400,000 / 47 |
| Pre-treatment [4] % of total | | | | | | | | 2,123,000 / 61 [1] | 170,000 / 3 | 301,000 / 6 |
| Landfill % of total | | 2,256,000 / 79 | 2,549,000 / 98 | 34,789 / 5 | 56,715 / 17 | 1,934,197 / 100 | 3,773,818 / 98 | 118,000 / 3 [3] | 1,600,000 / 32 | 2,500,000 / 49 |

Left axis labels: *Subcategory: use* (for HAZARDOUS WASTE subcategories); *Subcategory: use* (for NON-HAZARDOUS WASTE subcategories).

In cases where cells are blank, the particular country does not use the excavated soil in the indicated category. [1] Landfill coverage and use in other constructions. [2] Processing and sorting. [3] Included in recycling. [4] Pre-treatment of waste means sorting or biological treatment of contaminated soil and results in a secondary waste which may be recycled or disposed (Swedish EPA, 2016). [5] Recycling of waste means their use as a construction material on or outside landfills, as landfill coverage, as backfilling material or as soil improvement in land farming (Swedish EPA, 2016). [6] Except for mining waste. The difference in the years presented for each country reflects the reporting requirements and systems of the ELGIP Working Countries. It is plausible that trends would emerge with more data.

Whilst protective legislative frameworks are needed in order to protect the environment and human health from hazardous excavated soil, forward thinking and clearer guidelines are required in order to move towards a more sustainable reuse of excavated soils. The best practice frameworks designed to promote the reuse of excavated soils highlight the fact that reusing excavated soil has potential economic, environmental and social benefits in comparison to traditional disposal alternatives, thereby providing incentives to landowners and developers.

A summary of the different approaches according to country-specific guidelines are provided in the Supplementary Information (SI) for France, Norway, Portugal, Slovenia, Sweden (ELGIP Working Group countries), and Australia, Canada, England/Wales and Switzerland. Table S1 shows an overview of current practices in several countries to promote the reuse of excavated soil, Table S2 lists the type of soil included in the regulatory frameworks, Table S3 presents the application domain for excavated soil and Table S4 gives reference values for contamination of soils. Of note is that the regulatory frameworks are voluntarily in most countries (only three out of nine are mandatory); the treatment of contaminated soils in order to reduce pollutant levels within the realm of reuse is not allowed (with the exception of Canada, England/Wales, France and Slovenia); excavated soils can be reused on- and off-site in most countries (seven out of nine), and the number of contaminants and their threshold values are very variable between countries.

*2.2. European ELGIP Working Group Countries*

2.2.1. France

In France, the reuse of excavated soil outside the excavation site is allowed [22]. A mandatory requirement is knowledge related to whether the excavation site is considered to be contaminated and, as such, falls under the national policy for contaminated sites [23]. If the excavation site is not contaminated, then excavated soils are defined as natural materials and can be used in earthmoving programs (i.e., between sites) if they satisfy geotechnical considerations. If the excavated site is contaminated, then the excavated soil can be treated on-site to remove contaminants. In addition, containment, to control the migration of pollutants using physical and mechanical (e.g., geomembrane, sheet pile, diaphragm wall) or hydraulic measures, is possible. However, containment can only be used in cases where there is no relevant treatment solution, when the containment solution does not pose a threat to health, and when the durability of the solution can be demonstrated.

If excavated soils are transported off-site, then they are considered as waste and the code of environment is followed. However, these soils can be improved and subsequently reused if the following three conditions are met: (1) the soil quality of the recipient site is maintained, i.e., chemical properties of the excavated soil are consistent with the geochemical background of the recipient site, (2) the quality of the water resources at the recipient site are maintained and its ecosystems are preserved, and (3) chemical features of excavated soils are compatible with the expected use at the recipient site. In order to ensure these mandatory conditions are met, three levels of assessment are used. The first level is based on threshold concentrations of chemical elements and chemical compounds. The second level can be used in cases where the threshold concentrations are not met, as thresholds are given for specific construction development activities (industrial, office, landscaping) and for road construction [24]. The third level is for cases where these threshold limits are not met, and a site-specific study must be carried out.

2.2.2. Norway

Norway has adopted the WFD, however, surplus excavated soil is considered, per definition, as a waste material. The Norwegian Environment Agency have recently produced a guideline document that covers the intermediate storage and final disposal of clean soil and stones [25]. The reuse of clean soils and stones requires an exemption from the Pollution Regulation, and this can only be granted by the Environment Agency. The following criteria

must be met in order that clean excavated soil and stones can be reused: (1) site development has been designed in such a way that it is not influenced by the availability of clean excavated soil or stones, (2) the amount of material to be reused is sufficient for use, and (3) the excavated materials have properties that render them suitable for reuse. Concentration threshold values exist that can be used to determine whether a soil is clean or not. In cases where concentrations are exceeded, the soil is deemed contaminated. In cases where the soil contains pollutants that do not have corresponding threshold values, a risk assessment must be carried out to determine the hazard the soil poses to the surrounding environment. There are no current guidelines that apply to the reuse of contaminated excavated soils; however, it is possible to apply for a permit for reuse provided the reuse complies with the above criteria and a risk assessment can demonstrate no environmental impact.

### 2.2.3. Portugal

The regulation of soils excavated during construction works is according to the General Regulation of Waste Management [26], which was transposed from the WFD. This does not include non-contaminated soils, classified in accordance with Ontario Standards [27], and other natural materials, provided they are used for construction purposes in the following ways [28]: (1) on-site, in their natural state, or (2) off-site, in works subject to licensing or prior notice by the responsible authorities, in the environmental and landscape recovery of mining and quarrying activities, in the final cover of landfills, or at a site licensed by the City Hall. In cases when excavated soil (clean or contaminated) are not reused on- or off-site, they are classified as waste and the definition of their codes in the List of Waste, established in the WFD.

### 2.2.4. Slovenia

Within Slovenia, the management of excavated soils follows several laws (Law on Construction, Law on Mining, Law on the Protection of the Environment and Law on Waters), as well as the Regulation of the Waste [29], which is in line with the WFD, the Regulation on the Management of Waste arising from Construction Work [30] and the Regulation on Burdening of the Soil by Waste [31]. The Regulation on the Management of Waste arising from Construction Work stipulates requirements related to soil waste management. According to this regulation, the owner is fully responsible for construction waste management on-site. In cases where the excavated soil is not contaminated, the owner may reuse it on-site or off-site where owned by the owner.

The Regulation on Burdening of the Soil by Waste defines excavated soil as waste and presents physicochemical properties and permissible concentration limits for the following reuse scenarios: soil recultivation, backfilling of agricultural land, backfilling of building land and backfilling after excavation. The preparation of the excavation for its reuse is considered as waste recovery and requires an Environmental Permit from the Slovenian Environmental Agency. This permit is granted if sufficient documentation is provided, and the permissible concentration levels are met.

### 2.2.5. Sweden

Sweden has also adopted the WFD in Swedish Law. According to the Swedish Environmental Code (1998:808), excavated soils are classified as waste if the definition of waste is fulfilled or if the soil is not classified as a by-product. The Swedish Environmental Protection Agency [32] states that soil is not regarded as waste if it is excavated and used at the site where the excavation was carried out within a reasonable period of time (which is not numerically defined). If, however, a clear use for the soil does not exist, the soil must be classified as waste. Recycling of waste for construction purposes is allowed if the following prerequisites are fulfilled (SEPA, 2010): (1) the waste will replace traditional construction materials, (2) only the quantity needed for the construction shall be allowed, and (3) the construction shall fulfil a function. The recycling of non-hazardous waste for construction purposes requires a notification or a permit if the risk of pollution is low or more than

low according to 34-35 §§ in chapter 29 of the Environmental Regulation (2013:251). The SEPA provide threshold concentrations for some substances, which can be used in order to identify when the risk of pollution is less than low as described in chapter 29 of the Environmental Regulation (SEPA, 2010). However, recycling of waste when the risk of pollution is less than low may be governed by general rules in the future. The use of soils which are considered as a product (not waste) in construction works is regulated by laws such as the product legislation (Regulation (EC) N. 1907/2006 (REACH) and N. 1272/2008 (CLP)) and the rules of consideration in the Swedish Environmental Code (1998:808).

*2.3. Other European Associated Countries*

Switzerland and England/Wales have prepared guidelines to follow when excavated soil is reused. The Swiss guideline [20] is mandatory and serves the purpose of avoiding secondary pollution of the soil, groundwater and surface waters as a result of the deployment and reuse of contaminated soils. In addition, the guideline is only related to the pollutant content of excavated soils, and not physical aspects such as excavation, temporary storage or use of the excavated soil in landscaping. Pollutant concentrations are given in the guideline and the excavated soils to be reused should not exceed the threshold values. If the soils are to be reused at a different site, then the level of pollution in the soil at the new site must also be determined. Three so called "impact categories" are defined in the guideline as: (1) uncontaminated excavated soil, (2) weakly contaminated excavated soil, and (3) heavily contaminated excavated soil. Soils are placed in these categories based on the total concentration of pollutants. Uncontaminated excavated soils can be reused freely, weakly contaminated soil can only be reused on-site or in the close vicinity, and heavily contaminated soil cannot be reused.

The English/Welsh guideline document [16] is voluntary and provides four factors that should be considered when deciding whether excavated soils are waste materials: (1) protection of human health and protection of the environment, (2) suitability for use, (3) certainty of use, and (4) quantity of material in accordance with the principles of the WFD. There are three reuse scenarios that the English/Welsh guideline can be applied to: (1) Site of Origin, (2) Direct Transfer, and (3) Cluster Project. In the Site of Origin scenario, the excavated materials are reused at the same site from which they were excavated, either with on-site treatment, or without treatment. In the Direct Transfer scenario, clean, naturally occurring excess soils, can be transferred from one site to another for a reuse purpose. In this scenario, no soil treatment is involved, and soil is deemed suitable for reuse by meeting the beneficial reuse material specifications of the receiver site. The Cluster Project approach facilitates the remediation and/or development of a number of sites that are located in relatively close proximity and share a decontamination/treatment facility located at an authorized single site—the Hub site. A key principle of a Cluster Project is that the activity is temporary, and soils are passed between sites and to the Hub site for treatment and then reused.

*2.4. Non-European Countries*

Australia and Canada also have systems in place to ensure environmental protection whilst promoting the reuse of excavated soils. In New South Wales, Australia, the excavated natural material order of 2014 (New South Wales Environment Protection Agency, 2014) is a mandatory document and must be followed by those supplying excavated natural materials (i.e., soils). The document contains sampling requirements, concentration thresholds and methods that must be used to test the soils. Provided all of the requirements laid out in the document are fulfilled, then the excavated soil can be supplied to those who require it. In Ontario, Canada, a voluntary guideline [19] has been adopted which draws heavily on the English/Welsh guideline. The same principles are used to decide whether the soils are waste materials, and then two different routes are identified depending on whether the soil (considered as a non-waste) is intended for reuse in site remediation (contaminated soil) or site redevelopment (clean soil).

## 3. Barriers to the Reuse of Excavated Soil

Despite the increased interest in progressing towards reusing excavated soils, there are still several barriers that limit this practice. The barriers can be divided in to the following categories: (1) regulatory, (2) organizational (project planning and management), (3) logistical, and (4) material quality, which are characterized in the following four sections and summarized in Table 2. The barriers are not listed in an order of priority as this varies considerably depending on ELGIP Working Group country and on the group represented (those building on the ground, developers, consultants and policy makers). The barriers are reflective of the opinions of those who were involved in the data collection at the time of carrying out the work. Based on recent literature [33,34] and experience in the ELGIP Working Group countries, it is clear that there is no consensus on the best strategies to manage excavated soils in urban areas.

**Table 2.** Summary of barriers that exist for the reuse of excavated soils.

| Regulatory Barriers | Organizational Barriers | Logistical and Economic Barriers | Material Quality Barriers |
| --- | --- | --- | --- |
| Complicated legislation/regulatory framework that can include both local, regional and national governments/authorities. | Lack of knowledge and understanding of relevant policy and its application during construction works. | The supply and demand for excavated soil is not always inline (both spatially and temporally). | Rigid geotechnical requirements for soils that are to be reused (e.g., standards for construction materials). |
| Lack of guidelines for reuse in most countries. | Lack of holistic and early planning for possible reuse (preparation of applications, synergies with other projects, etc.). | Lack of intermediate storage capacity both on- and off-site. | Uncertainty of environmental risk related to reuse of lightly contaminated soil. Results in public resistance to reuse. |
| Long application/permit processing time when reuse is a possible option. | Contracts are not designed to promote reuse of excavated soil. | Limited permitted intermediate storage time for excavated soil. | Uncertainty about the quality of improved soil. Results in public resistance to reuse. |
| Ownership of reused soil and related risk-responsibility for potential future impacts. | | Extra cost for each logistical step (transport to off-site storage, etc.). | Lack of technical and accepted protocols to show compliance to technical specifications and legislation. |
| | | Reuse carries no economic incentive when compared to other solutions (e.g., landfilling). | Preference for virgin materials. |

### 3.1. Regulatory Barriers

Regulatory barriers are defined here as those that arise as a result of existing (or lack of) regulations from environmental authorities and other regulative bodies. A previous study by Ajayi and Oyedele [35] explored industry practitioners' viewpoints on effective policies for minimizing waste landfilled by the UK construction industry. The study relied on focus groups and questionnaires to gather data and concluded that increasing the stringency of regulatory tools remained the most viable way of stimulating waste diversion from landfill. Further afield in China, Jin et al. [36] investigated the production and disposal of construction and demolition waste as well as field practitioners' perceptions towards benefits and challenges of recycling and reuse. The authors concluded that a lack of supervision and regulations controlling construction and demolition waste recycle and reuse was the second most important barrier within the Chinese market. In addition, Menegaki and Damigos [37] report that law enforcement is one of the most critical drivers for successful management of waste from the construction industry. For project developers and contractors, the complicated and unclear legislation as well as the long application processing time are perceived as the biggest barriers to reuse. In order to reuse soil (or waste, in cases when such a definition is applied), several laws and regulations must be

followed. As demonstrated above for the five ELGIP countries, this is not harmonized and often results in a complicated process for both authorities and developers. For example, the WFD must be followed in cases where the soil is removed from a site as waste, whereas the Planning and Building Act and the Pollution Act should be followed in reuse scenarios. The sheer lack of mandatory guidelines and systems in place for reuse of soil from building and construction projects severely hampers efforts.

A permit from the environmental authorities is most often required in order to reuse excavated soil, and a risk assessment should be carried out to show that the environment and human health will not be compromised. Currently, there are a lack of guidelines for the overall permit application procedure and required documentation (e.g., soil characterization, risk assessment) which results in a great deal of uncertainty and reluctance to apply. In addition, long application processing times result in a barrier for projects that have strict time limitations. A further limitation is related to ownership of the material to be reused. The developer or landowner who reuses the soil is solely responsible for any future environmental impact of the new construction. However, if the developer or landowner dispose of the soil at a landfill, then they are freed of any subsequent liability questions. This, in itself, makes reuse a less attractive option.

### 3.2. Organizational Barriers

The main organizational barrier to the reuse of excavated soil is that the project planning process, in its current form, simply does not consider reuse of soil in most countries, even in the most developed ones. Without introducing new steps that open up for the reuse of excavated soil, it is difficult to assess whether this is a viable solution. It is most common, that in construction and demolition projects, the developer is responsible for the design and overall project implementation plan, and this includes the handling of excavated soils, while the contractor's role is limited to construction only. Formatting contracts in such a way means that a discussion related to reuse of excavated soils falls outside all parties' responsibility. The severe lack of a holistic approach also hampers to the reuse of excavated soils. The lack of consideration of project synergy, timescales and the movement of excavated soils from site to site to meet supply and demand, also hampers the degree of reuse [38].

In recent years in certain countries (e.g., Norway and Portugal), the use of the design-build contract form has increased in popularity. Here, the contractor is responsible for both design and construction, and they are the ones responsible for making decisions about mass flows and management. Design is then carried out by the contractor and begins at a later stage compared to a traditional build contract, where the developer plans the design before the contractor is engaged. In the case of the design-build contract form, an application for soil reuse would be prepared in the design stage by the contractor and may postpone construction work. This is incentive, in itself, to prepare the application as early as possible.

### 3.3. Logistical and Economic Barriers

The most significant logistical barrier for the reuse of excavated soil is that both temporal and spatial supply and demand does not often match. Thus, a problem arises related to the storage space needed for excavated soils in the interim before subsequent use. In many construction projects, on-site space is extremely limited, and excavated soils must be removed to temporary storage sites [39]. This is especially true in densely populated cities [40]. In addition to space, time is also a constraint as excavated soil is only permitted to be stored on- and off-site for a certain amount of time before its next use. According to the European Landfill Directive [41], waste can only be stored for 1 year as intermediate storage and for 3 years prior to recycling. Thus, storage longer than 1 year requires permission from authorities, and this obstacle must be overcome in order to fully exploit possible synergies between projects with a surplus of excavated soil and others that requires soil for reuse purposes, which can take place more than a year apart.

Logistical barriers are further amplified through additional transportation steps such as transport to and from temporary storage sites or to facilities where it is to be characterized or improved before being transferred back to the site where it is to be reused. This introduces additional direct transport and loading costs, and indirect costs such as the emission of greenhouse gases which combined may result in a net negative environmental and financial situation [42]. This financial barrier is heightened if the cost of virgin soil and landfill tax is low as there is no overall incentive for reuse. The economic barrier, whereby the positive or negative effect of reuse of excavated soil is felt, will inherently vary depending on country, and previous literature in this area is in its infancy. In a detailed study in Vietnam, increased reuse of waste from construction and demolition projects was postulated to be possible, provided aggressive and integrated strategies were developed, a key component of these being an evaluation of economic feasibility [43].

### 3.4. Material Quality Barriers

Excavated soils can be contaminated depending on their origin and/or historic usage. Therefore, their degree of contamination must be well characterized and assessed as a first step for reuse. In addition, these soils may have geotechnical variations, and must therefore be geotechnically characterized in order to evaluate whether the soil in question can be reused for the proposed purpose. Currently, there are still few countries with guidelines that specify which tests and risk assessment procedures should be carried out in order to document quality, which complicates matters [44]. A flowchart to screen excavated soils for several reuse strategies based on their current geotechnical and environmental properties has been proposed for a Brazilian situation, however, suggested to be fitting for worldwide use [45]. In addition to this, there are no specific guidelines in any of the countries in question that stipulate what documentation is needed to show an excavated soil meets geotechnical and geoenviromental property criteria that allow reuse. Given the documentation challenges above, as well as the other barriers discussed, the easiest way to satisfy the geotechnical and geoenvironmental requirements of civil engineering works is to use virgin material with well-known properties, rather than reusing excavated soils with a potentially inferior quality and hazardousness.

## 4. Ways to Increase the Reuse of Excavated Soil

It is of paramount importance that the reuse of excavated soils increases in order to take steps towards fulfilling the UN Sustainability Development Goal 11. Gains in the form of reduction of cost and climate impact via the reduction of transportation, landfilling and use of virgin materials can be made. One previous comprehensive study placed monetary and non-monetary values on benefits of reusing excavated soils and rocks for different scenarios [39]. Using studies from Eras et al. [5] and Chittoori et al. [45] presenting an industrial construction project and a pipeline construction project, Magnusson et al. [39], concluded that reusing excavated soil and rock on site could save up to 85% in terms of climate impact. In a further scenario based on data from CL:AIRE [46], the benefits of using excavated soil and rock in other projects was considered to be able to result in a cost saving of 30% and an emissions reduction of around 100 tons of $CO_2$. In the final scenario, recycling at facility, Magnusson et al. [39] looked in to the effect of excavated soil and rock that is classified as a waste being transported to a recycling facility, treated and prepared, then reused in other construction projects. Based on the work by Blengini and Garbarino [47], a conclusion was drawn that around 14 kg $CO_2$ equivalents per ton could be saved when excavated soil and rock was used compared to virgin materials.

This section provides some suggestions related to ways in which stakeholders (regulators, project planners and the construction sector) could increase the reuse of excavated soil. Based on the literature, discussions and previous experience, it is clear that a paradigm shift is required so that stakeholders see the benefits of the reuse of excavated soil and have correct decision-making tools. These suggestions, detailed below, are by no means exhaustive, and they represent the views of the authors and not necessarily stakeholders.

Regulation should be improved or developed to provide project planners and constructors with the tools they need to consider reuse. Harmonizing management systems within the same country, or across EU, will provide a greater understanding of the reuse of excavated soil. If individual municipalities in the same country operate with the same set of premises, then collaboration will be fostered and ultimately, expertise will be improved. Clearer and less complicated guidelines are needed that inform those considering reuse about how this can practically be achieved. In cases where the price difference with virgin material is not a sufficient incentive to drive reuse, landfill tax could be imposed. Another mechanism could be an evaluation system that rewards those who propose greater degrees of reuse and emphasizes their green construction practices.

A timelier planning process is required to optimize reuse. By focusing on the reuse of excavated soils earlier in the process, their management can be included, and environmental monitoring programs designed to ensure long-term questions can be answered. By identifying demand and availability of excavated soils in construction and demolition projects early in the planning process, reuse could be increased [48]. A significant step forward would be made if national reuse levels were stipulated as targets for individual projects. This would provide something for the planners and constructers to focus on when designing solutions. Combined with this, changing the way that tenders and subsequent contracts for major construction projects are organized to include requirements for how much excavated soil is reused, could increase the level of reuse.

Digital logistics systems that document excavated soil amounts and properties, sites that have a supply and demand, and type of reuse are tools that enable better project coordination. Traceability and quality control of the excavated soil with clear responsibilities are crucial. In Australia [18], Canada [19], England/Wales [16] and France [17], systems are in place that support traceability of excavated soil. To circumvent the problem related to lack of storage space on- and off-site, soil hubs (also called soil hotels) could be more widely implemented (e.g., [49]). These provide a more transparent market that regulates supply and demand for excavated soil. In several countries, the excavated soil is stored and then sold to projects with deficits. Regional and local mass handling plans could be developed or extended to accommodate the reuse of soil by taking a more holistic approach by covering a larger spatial scale. These plans may be able to identify areas that could be used as temporary storage (and processing) facilities. Careful selection based on proximity to developing urban areas will reduce transport costs, carbon dioxide emissions, and time pressures in building and construction projects. The plans could also include details of ways in which the excavated soil could be moved between sites.

Documentation has a central role to play in increasing the reuse of excavated soil. Documents must demonstrate suitable material quality which is fit for the intended reuse. Without greater demands on the documentation to show that the soil has acceptable geotechnical and geoenvironmental properties to be reused, virgin materials will be the preferred option. In order to provide assurance that the soil is fit for reuse, selected parameters and standard tests need to be identified and integrated with the intended reuse strategy [10,50,51].

## 5. Case Studies of the Reuse of Excavated Soil in ELGIP Countries

The ELGIP Working Group countries all have examples of successful reuse of excavated soil, and they are described below. It is hoped that by profiling these good examples of practical applications, in combination with the suggestions presented in the previous sections, reuse of excavated soil can be increased in the future.

In France, during the reshaping of the A36 motorway in the town of Mulhouse, an unusual dike of soil was found in the direct vicinity of the La Doller river. Its origin was unknown, but visual observations revealed that it was a mixture of natural soils (alluvial gravel and sand-mud-gravel) and CDW. In order to assess whether this mixture of materials, which had a volume of about 20,000 $m^3$, could be reused, 25 samples were chemically analyzed according to current legislation [52]. Results showed concentrations of

lead and antimony that exceeded the possibility of reuse without restriction, but permitted its reuse in road construction. However, based on proximity of the river, the decision-making process has led to the material being reused in landscaping, and covered with a geomembrane and clean top soil to avoid contact with surface water and groundwater.

In 2014, the Asak landfill, located in the Sørum municipality, 30 km north of Oslo (Norway), was opened to receive 1.5 million $m^3$ of clean and lightly contaminated CDW (excavated soil and concrete), which are being used in the improvement and construction of 600 m of a road running through Sørum, planned to be completed in 2020/21 [53]. The landfill is unique in this sense and uses those materials for a useful purpose. All of the financing needed to construct the road comes from landfill fees and the project provides a way to reuse excavated soils and to recycle concrete aggregates, as well as providing societal value in the form of a new road. The project requires a great deal of excavated soil and concrete to fill the topography according to the adopted project plan. The result is the transformation of low-quality agricultural land and an old, unsafe county road into an area with increased agricultural quality and an upgraded modern road. The road is constructed according to the Norwegian Road Authority specifications and the design of the landfill is in accordance with the Norwegian and EC landfill regulations.

In 2001, about 65,000 $m^3$ of fly ash released by electric arc furnaces, classified as non-hazardous waste, was transported to the waste rock spoils of the old coal mine of São Pedro da Cova, which is located near the city of Porto (Portugal). This recovery operation intended to fill the large existing cavities in the waste rock spoils and to create a horizontal platform with an area of about 11,000 $m^2$, where residential buildings were due to be constructed. However, a study performed in 2010 showed that the fly ash was hazardous waste [54]. This classification, in combination with the lack of containment structures to isolate the hazardous fly ash waste from the surrounding environment, as well as local site-specific conditions, showed that the hazardous fly ash waste posed a high risk to public health and the environment. In 2014–2015, the hazardous fly ash waste was transported to a hazardous waste landfill. Before removing the fly ash, 42,000 $m^3$ of clean soil which was on top of the hazardous fly ash waste was excavated and stored in local temporary deposits, in accordance with current Portuguese law practice [55]. This excavated soil was further backfilled on-site. This reuse of excavated soil on-site avoided the transportation of soils to an inert waste landfill, reduced carbon dioxide emissions, and allowed recovery of the landscape site, with relevant environmental, economic and social benefits.

In Slovenia, between the city of Maribor and the Slovenian-Hungarian border, 84 km of road was constructed. Aggregates ($8 \times 10^6$ $m^3$) were needed for the construction of key infrastructure, of which more than $1 \times 10^6$ $m^3$ was used for the base and sub-base pavement layers. During construction, obtaining the necessary aggregates with the correct geoenvironmental and geotechnical properties at an acceptable cost, was highlighted as a problem. To address this, a study was carried out to assess the technical feasibility of reusing the excavated soil produced during the earthworks for the construction of a highway. The soil was classified as gravelly soil with sand (maximum grain-size equal to 40 mm according to the Unified Soil Classification System as well-graded). Cyclic load tests provided information on the dynamic properties of the gravelly soil, which are often the cause of premature damage or even collapse of the road pavements. The results obtained showed that the gravelly soil was suitable for use in the embankments, and base and sub-base pavement layers. In order to reach optimal behavior of the base and sub-base pavement layers and a longer life cycle of the road, a mixture of 60% of crushed gravelly soil and 40% of uncrushed gravelly soil was used.

A new concrete area for plane parking was needed at Malmö/Sturup Airport (Sweden) in order to increase capacity. The designated area consisted of peat soil (37,000 $m^3$) that needed to be replaced by a different material. The fill material that was originally earmarked for this was 80,000 $m^3$ of reused masses of fine-grained till from the construction site. However, during test pit excavations, it was discovered that the natural water content of the fine-grained till was too high to allow the material to be used in its natural state

(Moisture Condition Value, MCV, was between 3 and 5, and it should be at least 7 to meet the requirements), and its geotechnical properties needed to be improved via a soil modification. Soil modification using cement and cement/lime mixtures were tested in laboratory and field. Soil compaction properties were determined according to the Moisture Condition Value (MCV) method via the use of unconfined compressive tests, and in the field, vane tests were used. Following modification, the moisture condition value increased 2 to 3 times, but the material strength increased nearly three-fold after a period of 24 h. Field tests were also carried out where the cement and cement/lime mixtures were added to the fine-grained till soil in the embankment and compacted with a smooth drum roller. After a curing period of 16 h, the undrained shear strength of the treated soil was improved and varied from 185 to 350 kPa. The results from this field test show the great potential that this modification technique had at the fill site [56], and the technical feasibility of reusing fine-grained till after its improvement.

## 6. Conclusions

Given the fact that the EU is striving towards becoming a smart, sustainable and inclusive economy by 2020, increasing the reuse of excavated soil is a necessity. Reusing excavated soils in a manner that does not pose a risk to the environment or human health could take a step towards more sustainable engineering practices and provide economic, environmental and social benefits. However, in Europe and in many other countries, reuse of excavated soil is, at present, extremely limited and often ends up at landfills. This paper has shown that there is a lack of common policy and practice surrounding the reuse of excavated soils from construction and demolition projects in the investigated countries. In fact, as highlighted by this work, there are many different approaches that can be taken with regards to the reuse of excavated soils. There is almost no overlap between practice or policy, and this suggest that a legal-wide framework is necessary to prevent soil degradation and ensure an adequate level of protection for all soils.

The possible solutions, outlined in this paper, to overcome the barriers that exist and could take steps towards increasing the reuse of excavated soils belong to the following categories: (1) regulatory (harmonized management systems, clear and simple guidelines and landfill tax), (2) organizational (early planning, stipulating the amount of soil to be targeted for reuse and designing contracts to promote reuse), (3) logistical (digital systems and establishment of soil hubs), and (4) quality (technical report ensuring that the soil is compatible with the new application). Follow-up work will be carried out to take steps towards suggesting a more harmonious framework for broader adoption.

**Supplementary Materials:** The following are available online at https://www.mdpi.com/article/10.3390/su13116083/s1, Table S1: Overview of current practices in several countries to promote the reuse of excavated soil, Table S2: Type of soil included in the regulatory frameworks, Table S3: Application domain for excavated soil, Table S4. Reference values for contamination of soils.

**Author Contributions:** Conceptualization: S.E.H., A.J.R., G.O., E.S., T.L., C.C., D.K., P.F., and B.Ž.; methodology: A.J.R., G.O., E.S., T.L., C.C., D.K., and B.Ž.; resources, S.E.H., A.J.R., G.O., E.S., T.L., C.C., D.K., P.F., and B.Ž.; writing—original draft preparation, S.E.H. and A.J.R., writing—review and editing, S.E.H., A.J.R., G.O., E.S., T.L., C.C., D.K., P.F., and B.Ž.; funding acquisition, G.O. and T.L. All authors have read and agreed to the published version of the manuscript.

**Funding:** This research was funded by the NGI Strategic Research Program, Geomaterials in a Circular Economy (GEOreCIRC), funded by the Norwegian Research Council, and the French Ministry of Environment.

**Institutional Review Board Statement:** Not applicable.

**Informed Consent Statement:** Not applicable.

**Data Availability Statement:** The data presented in this study are available on request from the corresponding author. The data are not publicly available due to the nature in which they were collected and the fact that the publication reflects the opinions of the authors only.

**Conflicts of Interest:** The authors declare no conflict of interest.

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
