# Peer review of "The Reuse of Excavated Soils from Construction and Demolition Projects: Limitations and Possibilities"

_sustainability, doi:10.3390/su13116083_

Round 1

Reviewer 1 Report

General comments

The manuscript "The reuse of excavated soils from construction and demolition projects: limitations and possibilities" deals with option and give insight into current problems of excavated soil. In the submitted article, it has been showing the possibility of soil utilization and discusses the related problems. The study is reporting on a highly relevant topic and that is of interest to a wider scientific audience. The original idea is interesting and appears that the manuscript has been successful in developing and verifying the aim of the study because the study and included several countries and a case study was used to verify the elaborated objectives. However, the case studies seem inconsistent with contrasting examples and without following a similar pattern of description. In general, paper can add merit to current knowledge and adheres to the journal’s standards in terms of language.

Specific comments

Line 34-35: Please add the area that refers to. Is it EU, Word or other?

Line 47-48: ...highlights the fact that sending clean excavated soil to landfill ..“ This sentence is unclear as the term clear soil has to be explained first (meaning). This could refer to natural, undisturbed or arable land so these different land uses can vary in soil quality and value as an asset. Therefore please use more appropriate phrases to indicate and depict soil that has been referring to.

Line 75: “Given the imperative need to implement sustainable practices in construction projects and the lack of reuse of...“ – I would suggest that it is not a lack of reuse but a lack of legislation/regulation

Line 99-103: Some additional information about the diverse workshop could add value to collected data

Line 106-107: Table 1 provides an overview of amounts of excavated soil that is landfilled. I would suggest that while discussing these issues keep in mind that those countries, among others characteristic, differ in territory and area.  Please incorporate that aspect. In addition to that in Table 1 different years are presented how this affects the summary results? This material can vary in time…

Line 127-128> This information was already presented in lines 71-72

Line 131: …. “developers. “ Maybe to clarify this noun in the introduction section who they are and what they do

Line 256-257: Please rewrite this sentence it has not a clear meaning.

Line 328: Table 2 -  it would be beneficial if listed barriers are organized in some hierarchical order to prioritize the list

              "Lack of focus and understanding of whether reuse is a viable solution. " this is not an organizational barrier- this is more sociological

              Instead of "quality barriers" insert "material quality barriers"

Line 375: What criteria were applied for the selection of case studies? In this section, only positive examples were presented but to get the whole impression the author could use a different approach to present a total number of excavation sites, and for instance, some indices of how to better present those good examples.

Line 396-407: This part of the manuscript deals with fly ash. Could this material consider as soil? If not please delete or replace with other

Line 421: " .....locally produced fine-grained till ". Please specify fine-grained till locally produced

Line 434-436: " Reusing excavated soils is a pillar of sustainable engineering and contributes to a reduction of carbon dioxide emissions (it is estimated to save up to 14 kg per ton) and to the costs of earthmoving (it is estimated that it can reach 435 85%)." These concluding remarks have not been properly discussed just data was repeated but not supported with findings

Line 440: This study focuses on EU so getting to the conclusion that the worldwide situation is the same could be an ambiguous statement without confirmation in the manuscript

Reviewer 2 Report

Dear authors,

Thank you for submitting an interesting manuscript. No supplementary information was available for download. However, reference was made in the manuscript.

Given the style of the manuscript, I anticipate that most information was collected in line with a panel discussion or expert meeting. However, no reference was made to how the data was compiled. Beyond this, there is a significant lack of references for statements made throughout the manuscript.

I recommend that a section is added that clearly states how information was compiled. In addition, I recommend that references are added throughout. Beyond this, a limitations section should be added.

Regards

Reviewer 3 Report

  Review of the manuscript on The Reuse of excavated soils----

 The manuscript   deals with the reuse of excavated soils and materials and describes the possibilities, opportunities, regulations for reuse, and limitations.  It is timely to emphasize the need of reuse of such materials considered as wastes or other types. The manuscript is well- written, and several case studies are provided.  The barriers provided for reuse are very valid, but I feel that economic barrier must also be included and described in this context, because in many countries it could be one of the serious barriers for recycling and reuse.  My other comments are rather minor.

L26: Add that after result show

L107.  Tons is an old unit and the new and SI unit is Mg (megagram) for weight. Please use this in the text and tables wherever, it applies.

 L208. Put, after exist.

L214. What implies with less than under low under SEPA regulations. It is not very clear. Rephrase the sentence.

 I did not find any supplementary tables here but suppose that they will appear in the published article.

Round 2

Reviewer 2 Report

Dear authors,

Thank you for providing an improved manuscript.

Please review the formatting - some headings are hard to identify.

I feel that the manuscript could have benefited from a separate short methods sections that briefly outlines data collection and potentially associated limitations.

Commonly, materials are referred to as soils to a depth of 2m -  construction projects often excavate materials from greater depths. Are you referring to those materials as soils?

I recommend adding clarification in this context.

Besides this, I can recommend your manuscript for publication.

All the best

Author Response

We thank the reviewer for their comments and have addressed them as detailed in the attachment.
